

# Tailoring transcranial alternating current stimulation based on endogenous event-related P3 to modulate premature responses: a feasibility study

Augusto J. Mendes[1,2,3], Alberto Lema[3], Sandra Carvalho[4] and Jorge Leite[5]

[1] Geneva Memory Center, Department of Rehabilitation and Geriatrics, University of Geneva, Geneva, Switzerland
[2] Laboratory of Neuroimaging of Aging (LANVIE), University of Geneva, Geneva, Switzerland
[3] Psychological Neuroscience Laboratory, CIPsi, School of Psychology, Universidade do Minho, Braga, Portugal
[4] Translational Neuropsychology Lab, Department of Education and Psychology, William James Center for Research (WJCR), University of Aveiro, Aveiro, Portugal
[5] CINTESIS@RISE, CINTESIS.UPT, Universidade Portucalense Infante D. Henrique, Porto, Portugal

Corresponding author
Jorge Leite, jorgel@upt.pt

## ABSTRACT

**Background**. Transcranial alternating current stimulation (tACS) is a brain stimulation method for modulating ongoing endogenous oscillatory activity at specified frequency during sensory and cognitive processes. Given the overlap between event-related potentials (ERPs) and event-related oscillations (EROs), ERPs can be studied as putative biomarkers of the effects of tACS in the brain during cognitive/sensory task performance.

**Objective**. This preliminary study aimed to test the feasibility of individually tailored tACS based on individual P3 (latency and frequency) elicited during a cued premature response task. Thus, tACS frequency was individually tailored to match target-P3 ERO for each participant. Likewise, the target onset in the task was adjusted to match the tACS phase and target-P3 latency.

**Methods**. Twelve healthy volunteers underwent tACS in two separate sessions while performing a premature response task. Target-P3 latency and ERO were calculated in a baseline block during the first session to allow a posterior synchronization between the tACS and the endogenous oscillatory activity. The cue and target-P3 amplitudes, delta/theta ERO, and power spectral density (PSD) were evaluated pre and post-tACS blocks.

**Results**. Target-P3 amplitude significantly increased after activetACS, when compared to sham. Evoked-delta during cue-P3 was decreased after tACS. No effects were found for delta ERO during target-P3 nor for the PSD and behavioral outcomes.

**Conclusion**. The present findings highlight the possible effect of phase synchronization between individualized tACS parameters and endogenous oscillatory activity, which may result in an enhancement of the underlying process (*i.e.*, an increase of target-P3). However, an unsuccessful synchronization between tACS and EEG activity might also result in a decrease in the evoked-delta activity during cue-P3. Further studies are needed to optimize the parameters of endogenous activity and tACS synchronization. The implications of the current results for future studies, including clinical studies, are

further discussed since transcranial alternating current stimulation can be individually tailored based on endogenous event-related P3 to modulate responses.

# INTRODUCTION

Transcranial alternating current stimulation (tACS) is a non-invasive method, in which, a weak electrical current is applied to the scalp at a certain frequency and intensity through two or more electrodes (*Herrmann et al., 2013*). Portions of this text were previously published as part of a preprint (https://hdl.handle.net/1822/80961) (*Mendes, 2022*).

The neuromodulatory effects of tACS have been observed *in vitro* (*Reato et al., 2010*) and *in vivo* studies (*Ali, Sellers & Fröhlich, 2013*; *Johnson et al., 2020*). Furthermore, recent clinical trials have suggested that repetitive sessions of tACS can induce changes in the brain's oscillatory activity that can outlast stimulation for a few weeks (*Ahn et al., 2019*; *Alexander et al., 2019*). These changes in brain oscillatory activity can be detected using event-related potentials (ERPs) due to a phenomenon called event-related oscillations (ERO) (*Herrmann et al., 2014*).

One of the most well-studied ERP components is the P3. This component is elicited in centroparietal regions, peaking between 250 and 600 ms after a relevant stimulus (*Polich, 2007*). The P3 elicited during an oddball paradigm (*i.e.,* pressing a button after an infrequent stimulus) is coupled with an increase in delta (0.5–4 Hz) and theta (4–7 Hz) activity, with spatial and temporal overlap of the P3 signal (*Güntekin & Başar, 2016*). Likewise, a recent study has shown the same concomitant activity after a target in a premature response paradigm, suggesting that the relation between delta activity and P3 is present in different cognitive tasks, namely in a target P3 that is elicited after a relevant stimulus that will result in a reward or punishment (*Mendes et al., 2024*). Apart from the target-P3, the analysis of the cue can be useful to determine a cue-P3 component, which is observed after a cue that precedes the target and has been interpreted as motivational attention towards the upcoming relevant-stimulus (*i.e.,* target) (*Broyd et al., 2012*).

The modulation of such ERP components can be useful as potential interventions targeting cognitive impairments related to impulsiveness. The evidence for the usefulness of the P3 component in assessing the effects of brain stimulation is not new. For instance, a recent meta-analysis showed that transcranial direct current stimulation (tDCS) applied in frontal areas can increase P3 amplitude in parietal sites during oddball and working memory tasks (*Mendes et al., 2022*). On the other hand, the use of EEG as a potential biological marker of response following tACS has also been suggested as a promising therapy (*Frohlich & Riddle, 2021*). In particular, the modulation of P3 amplitude holds significant potential in clinical conditions characterized by P3 abnormalities (*Kaiser et al., 2020*; *Pasion et al., 2018*; *Wada et al., 2019*). However, the precise impact of tACS on P3 amplitude remains unclear.

One study showed no significant effects of tACS on P3 amplitude and ERO during an oddball task when the tACS parameters were modified to match the oscillatory activity of each participant's P3 (more information about this procedure in Methods) (*Popp et al., 2019*). Another study, in an ADHD sample, showed a significant increase in the P3 amplitude following tailored tACS (but not in ERO), which was coupled with behavioral improvements (*Dallmer-Zerbe et al., 2020*). However, another study was unable to reproduce the previous results in a different sample of individuals with ADHD. In this study, the active tACS did not affect the P3 amplitude, but it did increase the N700 component compared to the sham stimulation (*Kannen et al., 2022*). A recent study with a similar methodology of delta tACS and P3 synchronization observed a concurrent increase in P3 amplitude and evoked delta activity during a visual task. (*Boetzel, Stecher & Herrmann, 2023*). Overall, the synchronization between tACS and endogenous EEG activity demonstrated promising results, even though mixed effects and limited sample sizes restrict our ability to draw definitive conclusions.

Moreover, there are also other studies testing tACS in EEG markers during cognitive tasks, although they do not employ the previously described synchronization methodology. For instance, in a recent study testing the effects of delta and theta tACS during a decision-making task, without synchronizing the ERO and P3 latency from each participant, the P3a and P3b amplitude decreased after the theta tACS when compared to sham (*Wischnewski, Alekseichuk & Schutter, 2021*). Likewise, another study that did not tune the parameters of tACS to the endogenous activity during a working memory task, has also shown inconsistent results (*Pahor & Jaušovec, 2018*). Namely, theta tACS applied bilaterally to the parietal region and at left fronto-parietal areas resulted in a decrease in resting-state theta band power, whereas no differences were observed in the P3 amplitude. On the other hand, the right fronto-parietal theta tACS did not affect theta power but increased the P3 amplitude during the working memory task (*Pahor & Jaušovec, 2018*). Together, these findings reveal distinct effects of the tACS on EEG outcomes that might be related to methodological differences such as the tuning of tACS parameters with ongoing oscillatory activity.

Therefore, the current study evaluates the feasibility and efficacy of phasic and frequency-tailored tACS in increasing target-P3 amplitude during a premature response paradigm. We implemented a tACS-EEG setup tested during an oddball paradigm (*Dallmer-Zerbe et al., 2020*; *Popp et al., 2019*), in which the ERO associated with the P3 is assessed for each participant to apply phasic and frequency tailored-tACS. The underlying assumption is that the synchronization between tACS and endogenous neuronal activity is extremely important to achieve better modulation of the oscillatory activity (*Riddle & Frohlich, 2021*). For that, the premature response task was adjusted to ensure the overlap between the peak of the sinusoidal tACS and the target-P3 peak. Likewise, the tACS frequency was also set based on the target-P3 ERO of each participant, whilst no synchronization between tACS and cue-P3 was employed. Hence, we hypothesize that active tailored tACS will increase the target-P3 amplitude in comparison with sham stimulation, while no effects are expected in the cue-P3 amplitude. Moreover, considering the expected modulations in target-P3 amplitude, we also hypothesize an increase in the ERO target-P3 activity (and

not in Cue-P3). Finally, we expect that the increased amplitude of the target-P3 will be coupled with improvements in the premature response paradigm.

## MATERIALS & METHODS

### Participants

Twelve healthy volunteers (seven females; mean age: $25.67 \pm 1.97$) participated in the study and signed the informed consent before their enrollment in the study. The participants were right-handed and without any history of neurological and psychiatric disorders. The study was approved by the University of Minho ethics committee (Ethics Committee for Research in Life and Health Sciences; CEICVS 057/2021) and we followed the recommendations of the Declaration of Helsinki regarding ethical standards and participants' safety.

### Study design

The study was performed in two sessions with distinct tACS conditions, namely active and sham stimulation. The order of the stimulations was counterbalanced and randomly assigned to the participants. Sessions were scheduled at least 48 h apart to avoid carryover effects. In the first session, self-report questionnaires were administered to evaluate handedness, impulsive traits, and clinical symptomatology (see Table S1). Additionally, in the first session, participants performed a baseline block of the Cued Premature Response Task (CPRT) concomitant with the EEG data collection, to assess the latency and the ERO associated with the P3 component. Afterwards, three blocks of the waiting impulsivity task were performed, in which the first and the third ones were performed only with EEG (*i.e.,* pre and post-tACS), while the second block was performed concomitantly with tACS (see Fig. 1A). The second session did not include the baseline block; thus, participants only performed the last three experimental blocks (*i.e.,* pre, during, and post-tACS). At the end of each session, participants completed a blinding questionnaire (see results in Table S2).

### Cued premature response task

The CPRT was programmed and executed in E-Prime 3 software (Psychology Software Tools, Pittsburgh, PA, USA) based on a pre-print by *Mendes et al. (2024)*. The baseline block performed only in the first session comprised 10 training trials and 100 experimental trials. Moreover, the following three experimental blocks also comprised 100 experimental trials each. The total duration of each block was approximately 10 min, totaling 40 min in the first session (*i.e.,* 30 min in the second session), including several pauses between the experimental blocks. Participants were asked to press a button in the E-Prime Chronos response box to begin the experiment, and to release it as soon as possible following the target appearance on screen. The target was always preceded by a cue, which informed the participant that the target was about to be displayed. In the first and last blocks (*i.e.,* pre and post-tACS), the interval of time between the start of the trial and the cue was jittered between 1000 and 1500 ms, while the interval between the cue and target was jittered between 500 and 2500 ms (see Fig. 1B). On the other hand, the block of CPRT during tACS had a "wait" adjustment that was dependent on the stimulation parameters that were individually estimated in the EEG online analysis of the baseline block (see

**(A)** *Experimental Design*

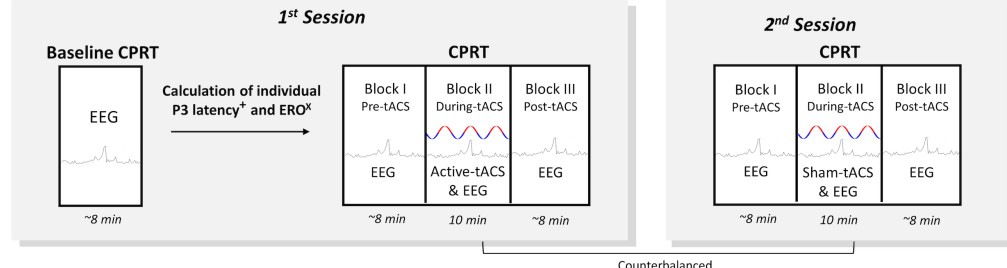

**(B)** *Cued Premature Response Task (CPRT)*

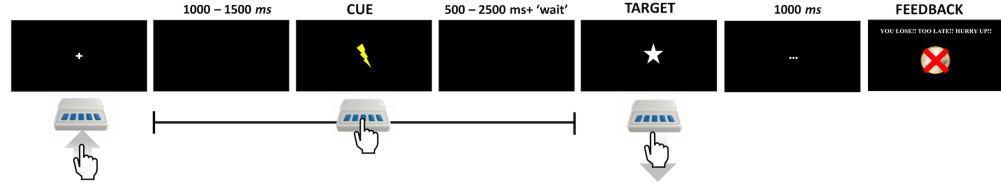

**(C)** *tACS-EEG Synchronization Setup*   **(D)** *tACS*

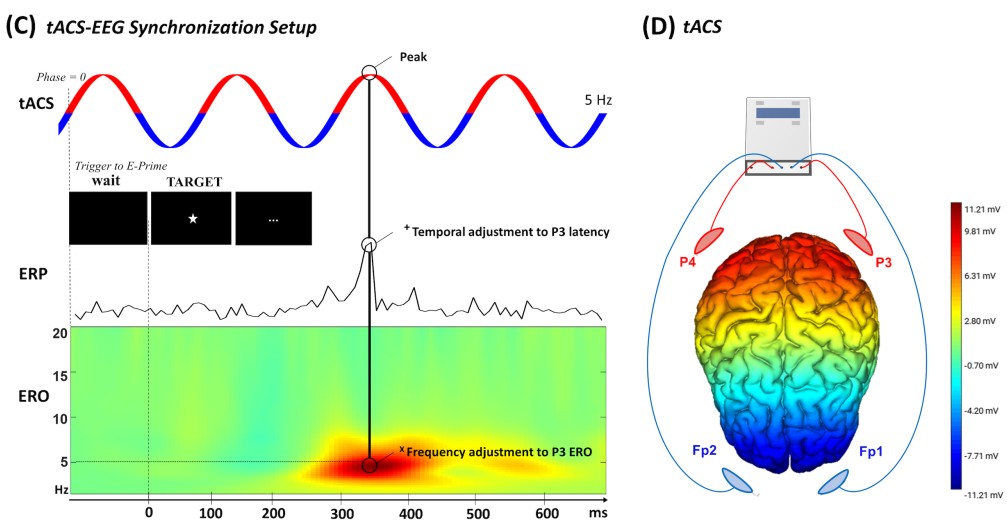

**Figure 1** **Overview of the study design.** (A) Overview of the study design with two distinct sessions. (B) The experimental task to evaluate premature responses. (C) The temporal adjustment synchronizing the P3 latency calculated in the baseline block with the tACS peak was accomplished with the inclusion of a tailored "waiting" period before the display of the target in the CPRT. On the other hand, the frequency of tACS was calculated based on the P3 ERO, which was the frequency with the maximum dB value within the P3 time-window. Finally, (D) tACS electrodes were placed in two electrode clusters (*i.e.,* P3 and P4 & Fp1 and Fp2) that were interchangeably anode and cathode (tACS figure is merely illustrative of the setup). The electric field map representing the voltage topography distribution when P3 and P4 electrodes deliver anodal stimulation and Fp1 and Fp2 electrodes deliver cathodal stimulation was computed in the NIC 2.0 software (Neuroelectrics, Barcelona, Spain).

'Electrophysiological acquisition and data analysis') (*Dallmer-Zerbe et al., 2020*). This is of particular interest since the tACS needs to match the temporal properties of the oscillatory activity. We aimed to entrain electrophysiological activity that occurred in a specific period after target onset (*Jones, 2016*). In this block, the interval between cue and target was

randomly selected between 300 and 2300 ms plus the "waiting" period (see Fig. 1C), to ensure the synchronization between the peak of tACS and the target P3 latency, thus minimizing "drifts" between the onset of stimulation and the stimulus presentation. This synchronization was controlled in MATLAB (MathWorks, Natick, MA, USA) with the MatNIC package to trigger the stimulation by the Starstim R20 (Neuroelectrics, Barcelona, Spain). MATLAB initiated stimulation by transmitting a UDP trigger to the Python console, which subsequently transmitted iterative TCP triggers to E-Prime 3 whenever the tACS reached phase 0 (please check the MATLAB and Python scripts with the E-prime file of CPRT in the Open Science Framework (OSF) repository). Please refer to Figs. S1 and S2 for some examples.

The reinforcement/punishment feedback pretended to elicit a higher number of premature responses. Therefore, the participant's response was rewarded with virtual money if his/her response was fast, punished if his/her response was slow, or neither rewarded nor punished if they released the button before the target onset (*i.e.,* premature response). The feedback was tailored for each participant based on the mean and variability of the response time (RT) observed in the last 10 trials of the baseline block. The mean and variability of the response time (RT) observed in the last 10 trials of the baseline block were considered to estimate the reinforcement/punishment feedback (see Fig. SM3), specifically:

- Very fast responses: if the participant released the button with an RT below −0.66 standard deviation (SD) of the baseline RT mean, the participant was reinforced with virtual 1€. Moreover, if any participant earned 1€ three times in a row, the feedback increased to 2€ for "very fast responses".
- Fast responses: if the participant released the button with an RT between −0.66 SD and +0.33 SD of the baseline RT mean, the participant earned a virtual 0.5€.
- Slow responses: if the participant released the button with an RT between +0.33 SD and +1SD of the baseline RT mean, the participant lost virtual 0.5€.
- Very slow responses: if the participant released the button with an RT above +1SD of the baseline RT mean, the participant lost virtual 1€.
- Premature responses: if a participant released the Chronos button before the target, the feedback was "Continue", in a way that participants were not reinforced nor punished.

## Transcranial alternating current stimulation

tACS conditions were applied in distinct sessions through the Starstim R20 (Neuroelectrics, Barcelona, Spain). The 25 cm$^2$ round saline-soaked electrode sponges (~radius of three cm, current density: 0.08 mA/cm2) were placed in two clusters of two electrodes each, specifically in the parietal areas (*i.e.,* P3 and P4) and supraorbital area (*i.e.,* Fp1 and Fp2) (see Fig. 1D). The clusters delivered alternatingly anodal and cathodal stimulation to ensure sinusoidal stimulation (*Popp et al., 2019*). The active tACS was delivered at 2 mA (peak-to-peak) intensity for 10 min (with 15 s of ramp up and ramp down) during the CPR Task. tACS frequency was individualized for each participant in the ranges of delta and theta band (1.5–7 Hz) according to the peak ERO power detected in the baseline block (check Table S3 for individual information). The average stimulation frequency was 3.29 Hz (SD = 1.9 Hz) in line with previous studies, using a similar methodology (*Dallmer-Zerbe*

*et al., 2020*). For the sham procedure, tACS was delivered for 15 s (with 15 s of ramp up and ramp down: 45 s in total) at the beginning and the end of the 10-minute session.

## Electrophysiological acquisition and data analysis

EEG data collection was performed with the Starstim R20 (Neuroelectrics, Barcelona, Spain) in the following electrodes: F7, F3, Fz, F4, F8, C3, Cz, C4, T7, P7, Pz, T8, P8, O1, Oz, and O2. Data was sampled at a rate of 500 Hz and analyzed using the EEGLAB toolbox (*Delorme & Makeig, 2004a*) at two specific times: online and offline analysis. The online analysis was performed between the baseline block and the first experimental block of the first session, whilst the offline comprehended the remaining EEG data, namely pre and post-tACS blocks from both sessions.

The EEG analysis started with filtering between 0.5 and 40 Hz using an FIR filter and the line noise (*i.e.,* 50 Hz) using a notch filter before re-referencing to the average. This continuous data was epoched around the cue and the target with a total duration of four seconds (*i.e.,* 2000 ms before and after the stimulus). A baseline correction considering the 200 ms before the target onset was performed. The cue epochs with a cue-target interval lower than 800 ms or with a premature response in the initial 1000 ms were removed from the analysis. Moreover, epochs exceeding $\pm 100\,\mu$V in the Cz or Pz during the time window of interest (*i.e.,* between $-200$ ms and 600 ms) were also removed. Finally, a visual inspection was also performed to detect potential artifacts not been removed before. The rejection rate was approximately 10% of the epochs, which followed the recommendations by *Delorme, Sejnowski & Makeig (2007)* (see Table S4).

The offline data analysis was similar to the online plus the artifact removal through the ICA. This step was only performed in the offline analysis considering the required computational time and comprehensive examination of each component. The grand averaged ERPs in the figures were filtered using a 12 Hz low-pass filter. One participant was removed from the data analysis because the post-tACS file of the sham session was corrupted.

### Event-related potential: P3

The online analysis calculated the target-P3 latency considering the peak on the time window between 250 and 600 ms in the Pz electrode, based on a previous study (*Mendes et al., 2024*). The latency was required to identify the maximum dB value in the online analysis (see ERO subsection). The offline analysis focused on cue and target-P3 amplitude in Pz. The P3 amplitude was calculated with previous time-windows used in literature (*Broyd et al., 2012*; *Mendes et al., 2024*), specifically the average amplitude between 250 and 450 ms for the target-P3 and between 350 and 600 ms for the cue-P3.

### Event-related oscillations

The ERO analysis was performed with the EEGLAB function *newtimef()* (*Delorme & Makeig, 2004a*). Specifically, 3-cycle Morlet wavelets were used for the time-frequency decomposition (*i.e.,* frequency resolution = 0.25 Hz; temporal resolution = 8 ms). The baseline normalization was chosen considering the additive model through an unbiased single-trial normalization method (*Grandchamp & Delorme, 2011*). The normalization

method subtracted to each epoch the average activity of the whole epoch. Then, the dB conversion was performed in each epoch considering the baseline period of 1000 ms before the target onset. In the online analysis, based on *Dallmer-Zerbe et al. (2020)*, the dB from the Pz electrode was averaged around ±150 ms the P3 latency for each participant, and the maximum value was identified as the P3 ERO and consequently the stimulation frequency to apply (*Broyd et al., 2012*) (see Fig. 1C). In the offline analysis, the delta (1.5–4 Hz) and theta (4–7 Hz) bands were averaged for the Pz electrode site, using the aforementioned time windows for cue and target-P3.

*Power spectral analysis*

The power spectral density (PSD) was analyzed using the function *spectopo()* from the EEGLAB toolbox using the fast Fourier transform (FFT) (*Delorme & Makeig, 2004a*). The delta and theta band power was estimated for Pz through the Welch method with a Hamming window (*i.e.,* window length: 500 points; FFT length: 500 points). Moreover, an overlap of 20% of the sampling rate (*i.e.,* 100 points) was set, to control for the leakage effect caused by epoching. The PSD was performed independently for cue and target-P3 epochs.

*Additional analysis*

We also performed an EEG analysis tailored to each participant based on the methods proposed by *Dallmer-Zerbe et al. (2020)*. Considering the endogenous electrophysiological activity found in the baseline block, three analyses were performed: (i) adjusted to the P3 latency of each participant, (ii) adjusted to the tACS frequency applied to each participant, and (iii) adjusted simultaneously to the P3 latency and tACS frequency. About the temporal adjustment (i), the ERP and ERO were averaged around ±150 ms relative to the P3 latency calculated in the online analysis. The temporal adjustment analysis was only performed in target-P3 epochs given that the latency was estimated only for target-P3 (not for cue-P3). Likewise, the frequency adjustment (ii) was performed in the ERO and PSD analysis by averaging around ±3 Hz relative to the tACS frequency of each participant (instead of averaging dB in delta and theta bands). Finally, the adjustment to the time window and to the frequency (iii) was a combination of both analyses explained before.

## Sample size calculation

The sample size calculation was based on an effect estimate of 1.2 (Cohen's *d*) between active and sham sessions observed in the preliminary study testing tACS in P3 amplitude (*Dallmer-Zerbe et al., 2020*). Thus, the sample size was estimated to find a within-group effect between active and sham tACS in P3 amplitude with a statistical power of 95% and an alpha level of 5%. A total of 10 participants were calculated, but we added two participants to the estimation considering potential differences with the aforementioned study. The final sample size for this feasibility study was 12 participants. The calculation was performed in G*Power version 3.1.9.7 (*Faul et al., 2009*).

## Statistical analysis

The absolute change was calculated between pre and post-tACS for each session (*i.e.,* active and sham). For that, the values observed in the block before tACS were subtracted from the

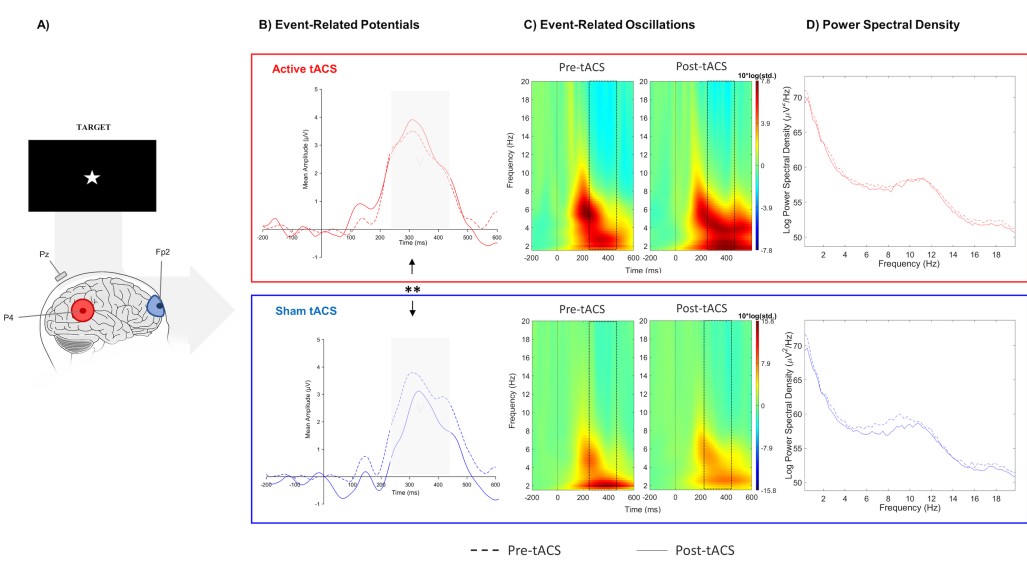

**Figure 2 Results from EEG analysis of target-P3 at Pz electrode site.** (A) Results from EEG analysis of target-P3 at Pz electrode. (B) Grand mean of the event-related potential in the time-window of interest (represented in the gray area and dashed lines: 250–450 ms), (C) event-related oscillations, and (D) power spectral density in pre- and post-tACS block in both sessions. ** $p < 0.01$ (significant effect between both tACS session in the P3 amplitude).

values obtained after tACS. This estimation was performed in every EEG and behavioral outcome. We followed a similar procedure to the one followed by *Dallmer-Zerbe et al. (2020)*. However, we chose the absolute change instead, because relative change may lead to increased variability when there are very low values in the pre-tACS block. Paired t-tests were used to assess the absolute change observed between active and sham sessions if the difference between both tACS conditions followed a normal distribution according to the Shapiro–Wilk test. Otherwise, non-parametric analysis was performed, specifically the Wilcoxon signed-rank test. A *p*-value of 0.05 in a two-tailed test (0.025 for each direction) was considered the significance threshold for all analyses. The statistics were analyzed in R (*R Core Team, 2020*).

# RESULTS

## Event-related potentials: P3

The nonparametric analysis revealed a significant difference in the absolute change between active and sham sessions in target-P3 amplitude ($W(10) = 63$, $p = 0.01$), but no significant effect was detected for the cue-P3 ($t(10) = -0.11$, $p = 0.91$). In the additional analysis considering the temporal adjustment around P3 latency to each participant (i), there was a significant effect in target-P3 amplitude between active and sham tACS ($t(10) = 2.84$, $p = 0.02$). The absolute change of target-P3 amplitude was higher in the active session in comparison with the sham in both analyses (Fig. 2B; Table 1).

**Table 1** Descriptive (mean and SD) and inferential statistics in ERP, ERO, and PSD analysis for target-P3.

| | | Active tACS | | | Sham tACS | | | $t / V$ | $p$-value |
|---|---|---|---|---|---|---|---|---|---|
| | | Pre | Post | Absolute Change | Pre | Post | Abs. change | | |
| Target- P3 (250–450 $ms$) | ERP (μV) | 3.45 (1.63) | 3.49 (1.15) | 0.04 (1.02) | 3.96 (1.17) | 2.83 (0.98) | −1.14 (0.96) | 63 | 0.005 |
| | Delta (dB) | 5.31 (4.99) | 6.62 (3.82) | 1.31 (4.16) | 8.29 (9.66) | 6.26 (3.99) | −2.03 (9.20) | 1.55 | 0.153 |
| | Theta (dB) | 3.75 (3.75) | 4.58 (3.69) | 0.83 (3.61) | 3.96 (5.73) | 4.47 (4.04) | 0.50 (5.53) | 0.26 | 0.797 |
| | Adjusted Frequency (dB) | 4.80 (4.01) | 5.76 (3.25) | 0.96 (3.01) | 6.86 (8.48) | 5.86 (3.59) | −1.01 (8.14) | 0.98 | 0.349 |
| Target-P3 (Adjusted time: ±150ms around P3 latency) | ERP (μV) | 2.86 (1.18) | 2.78 (1.16) | −0.09 (0.86) | 3.11 (0.89) | 2.08 (0.64) | −1.03 (0.95) | 2.84 | 0.017 |
| | Delta (dB) | 5.08 (4.56) | 6.22 (3.28) | 1.14 (3.66) | 7.71 (8.58) | 5.64 (3.73) | −2.06 (7.75) | 1.59 | 0.142 |
| | Theta (dB) | 4.02 (4.01) | 5.22 (4.17) | 1.20 (3.57) | 4.35 (5.76) | 4.28 (3.57) | −0.07 (5.44) | 0.79 | 0.448 |
| | Adjusted Frequency (dB) | 4.85 (3.60) | 6.01 (3.27) | 1.16 (3.17) | 6.71 (7.77) | 5.64 (3.73) | −1.07 (7.33) | 1.11 | 0.294 |
| Spectral Analysis | Delta (dB) | 7.62 (2.41) | 7.08 (2.34) | −0.54 (2.21) | 7.43 (2.27) | 6.81 (2.59) | −0.62 (2.27) | 37 | 0.765 |
| | Theta (dB) | 0.75 (2.17) | 0.43 (1.83) | −0.31 (1.48) | 1.24 (3.02) | 0.42 (2.21) | −0.82 (2.82) | 45 | 0.32 |
| | Adjusted Frequency (dB) | 5.09 (3.53) | 4.72 (3.84) | −0.39 (1.71) | 5.54 (4.26) | 4.22 (4.08) | −1.32 (3.25) | 0.73 | 0.484 |

**Notes.**
Absolute change is the subtraction of Post–Pre

### Event-related oscillations

No significant effects between both sessions were observed in evoked-delta (t(10) = 1.55, $p$ = 0.15) and evoked-theta (t(10) = 0.26, $p$ = 0.79) during target-P3 (Fig. 2C; Table 1). On the other hand, paired t-tests revealed a marginal significant effect in delta activity during cue-P3 (t(10) = −2.06, $p$ = 0.07), but not for theta (t(10) = −1.54, $p$ = 0.15) (Fig. 3C; Table 2). Furthermore, the additional analysis of the temporal adjustment (i) did not reveal significant differences in delta (t(10) = 1.59, $p$ = 0.14) and theta (t(10) = 0.79, $p$ = 0.45) during target-P3 between sessions. In the frequency adjustment analysis (ii), no significant differences were detected in the adjusted ERO from target-P3 (t(10) = 0.98, $p$ = 0.35), but, a significant effect between both tACS session in the adjusted ERO from cue-P3 (t(10) = −2.07, $p$ = 0.03) was found. The active tACS session showed a significant decrease in the adjusted ERO in comparison with the sham. Finally, the temporal and frequency analysis performed in target-P3 (iii) did not reveal a significant effect on adjusted ERO between active and sham (t(10) = 1.11, $p$ = 0.29).

### Power spectral analysis

No differences were found in terms of the absolute change between active and sham sessions for the delta (W(10) = 37, $p$ = 0.77) and theta bands (W(10) = 45, $p$ = 0.32) in target-P3 epochs (Fig. 2D; Table 1). Likewise, no significant differences in spectral power of cue-P3 epochs for delta (t(10) = 0.04, $p$ = 0.97) and theta (t(10) = 0.46, $p$ = 0.66) (Fig. 3D; Table 2) were found. Finally, regarding the frequency adjustment analysis (ii), there were no

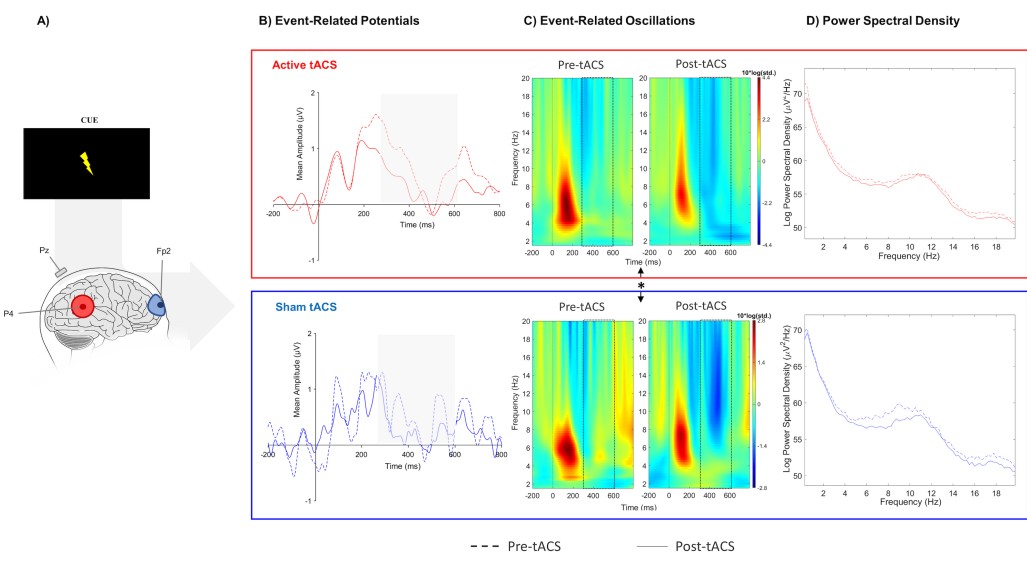

**Figure 3** **Results from EEG analysis of cue-P3 at Pz electrode site.** (A) Results from EEG analysis of cue-P3 at Pz electrode. (B) Grand mean of the event-related potentials in the time-window of interest (represented in the gray area and dashed lines: 350 –600 ms),(C) event-related oscillations, and (D) power spectral density in pre- and post-tACS block in both sessions. * $p < 0.05$ (significant effect between both tACS session in the adjusted ERO).

significant differences between both sessions for the target-P3 (t(10) = 0.73, $p = 0.48$) and the cue-P3 epochs (t(10) = 0.61, $p = 0.56$).

## Behavioral analysis

The paired $t$-test did not reveal statistically significant effects in the number of premature responses between the active and the sham session (t(11) = −0.51, $p = 0.62$). Likewise, the nonparametric analysis also did not show significant effects in the absolute change of total money earned/loss (W(11) = 41, $p = 0.51$) and release time (W(11) = 41, $p = 0.91$) between both sessions (Table 3).

## DISCUSSION

This preliminary study tested the feasibility of individualizing tACS delivery based on the individual P3 (latency and frequency), namely by matching tACS parameters to the target-P3 ERO estimated for each participant during a premature response paradigm. In the sham session, the amplitude decreased in the post-tACS block, whilst in the active session this decrease was not observed. The P3 amplitude typically decreases during a session due to habituation (*Polich, 1989*), but the tailored delta/theta tACS counteracted this expected decline. Nonetheless, no effects of tACS effects were found in the ERO analysis during the target-P3 time window. This is of particular interest given that a trend for a decrease of the ERO activity in the active session during cue-P3 was observed. Furthermore, tACS neither resulted in a broader oscillatory activity modulation as suggested by PSD analysis nor significantly changed any behavioral outcome, as assessed by the CPRT.

Mendes et al. (2024), *PeerJ*, DOI 10.7717/peerj.17144

**Table 2** Descriptive (mean and SD) and inferential statistics in ERP, ERO, and PSD analysis for cue-P3.

| | | Active tACS | | | Sham tACS | | | *t / V* | *p-value* |
|---|---|---|---|---|---|---|---|---|---|
| | | Pre | Post | Absolute Change | Pre | Post | Abs. Change | | |
| Cue- P3 (350–600 *ms*) | ERP (μV) | 0.40 (0.58) | 0.19 (1.41) | −0.20 (1.52) | 0.43 (1.05) | 0.29 (0.69) | −0.14 (1.28) | −0.11 | 0.912 |
| | Delta (dB) | −0.57 (1.19) | −1.19 (1.33) | −0.62 (1.65) | −0.31 (0.95) | 0.02 (0.81) | 0.33 (1.36) | −2.06 | 0.067 |
| | Theta (dB) | −0.29 (0.91) | −1.20 (0.63) | −0.91 (0.75) | 0.18 (0.93) | −0.34 (1.18) | −0.51 (1.20) | −1.54 | 0.154 |
| | Adjusted Frequency (dB) | −0.48 (1.13) | −1.05 (0.79) | −0.57 (1.20) | −0.19 (0.82) | −0.14 (0.96) | 0.06 (1.31) | −2.07 | 0.032 |
| Spectral Analysis | Delta (dB) | 7.47 (2.76) | 7.08 (2.34) | −0.39 (2.50) | 7.26 (2.14) | 6.81 (2.59) | −0.44 (2.22) | 0.04 | 0.972 |
| | Theta (dB) | 0.39 (2.48) | 0.43 (1.83) | 0.04 (1.79) | 0.94 (2.77) | 0.42 (2.21) | −0.51 (2.53) | 0.46 | 0.658 |
| | Adjusted Frequency (dB) | 4.94 (3.87) | 4.72 (4.84) | −0.22 (1.93) | 5.26 (4.02) | 4.22 (4.08) | −1.04 (3.08) | 0.61 | 0.556 |

**Notes.**
Absolute change is the subtraction of Post–Pre.

Mendes et al. (2024), *PeerJ*, DOI 10.7717/peerj.17144

**Table 3  Descriptive (mean and SD) and inferential statistics for CPRT outcomes.**

| | | Active tACS | | | | Sham tACS | | | | *t / V* | *p-value* |
|---|---|---|---|---|---|---|---|---|---|---|---|
| | | **Pre** | **During** | **Post** | **Abs. Change** | **Pre** | **During** | **Post** | **Abs. Change** | | |
| | Premature Responses | 17 (11.21) | 13.5 (8.94) | 16 (10.88) | −1 (7.64) | 16.67 (10.96) | 15.42 (10.5) | 17 (11.39) | 0.33 (5.12) | −0.51 | 0.615 |
| Cued Premature Response Task | Monetary Gain/Loss | 21.13 (38.02) | 13 (32.51) | 17.13 (29.17) | −4 (15.53) | 21.54 (30.61) | 20.13 (24.79) | 16.92 (25.92) | −4.63 (15.72) | 41 | 0.505 |
| | Release time *(ms)* | 189.27 (48.67) | 203.35 (45.89) | 193.51 (61.34) | 4.24 (27.69) | 192.58 (54.31) | 196.3 (49.47) | 187.17 (47.91) | −5.4 (36.99) | 41 | 0.91 |

**Notes.**

Absolute change is the subtraction of Post–Pre.

Our findings suggest the need for phase synchronization between the tACS and endogenous activity. For instance, the Arnold tongue effect suggests that the entrainment of neuronal oscillations is achieved with lower-intensity stimulation if they share the same frequency and phase (*Notbohm, Kurths & Herrmann, 2016*). This is of particular interest because matching the phase and frequency between tACS and P3 ERO was performed specifically for target-P3 (and not for the cue-P3), which was successfully increased in the active session. Consequently, this study also suggests that tACS effects may be limited to transient activity (*i.e.,* P3), instead of the broad oscillatory activity measured in the PSD. This notion is in line with recent work suggesting that neuronal oscillations may be rhythmic bursts instead of sustained oscillations (*Jones, 2016*; *Van Ede et al., 2018*). Thus, higher target-P3 amplitude observed during the active tACS session might be due to the increase of delta/theta bursts after the target, rather than an increase of sustained delta/theta activity (*Mendes et al., 2024*). Moreover, previous studies have already shown that theta tACS is capable of increasing the transient theta activity during cognitive tasks (*Hsu et al., 2017*; *Vosskuhl, Huster & Herrmann, 2015*); however, no effects of tACS were detected for the resting theta activity (*Mosbacher et al., 2021*; *Wischnewski & Schutter, 2017*). Nevertheless, our results failed to detect a statistically significant effect in delta activity during target-P3.

Even though there were no tACS effects on the cue-P3 amplitude modulation, a trend for a decrease in the evoked-delta (ERO) was observed during the cue-P3, which can potentially be explained by an anti-phasic effect between the stimulation and the evoked oscillation (see Fig. 4A). This statement is speculative because we did not synchronize tACS with cue-P3. However, it is worth noting that the opposite effect of tACS on EROs has been previously documented. For instance, the decrease in event-related delta activity during a cognitive task was already observed after the application of delta tACS (*Wischnewski & Schutter, 2017*). However, the authors of the previous study did not synchronize tACS and the underlying oscillatory activity, which can explain the unexpected effect (*i.e.,* decrease of evoked-delta) through the mismatch between both signals (*Wischnewski & Schutter, 2017*). So, it is possible, that this decrease in delta-activity could be explained by the spike-timing dependent plasticity (STDP) hypothesis, which proposes that tACS is most effective in oscillatory activity above the stimulation frequency (*Vogeti, Boetzel & Herrmann, 2022*; *Zaehle, Rach & Herrmann, 2010*). If the dominant oscillation between two neurons is higher than the stimulation frequency, there is a strengthening of the synapse (*i.e.,* long-term potentiation, LTP) because pre-synaptic events occur before the post-synaptic. On the other hand, if the stimulation frequency is higher than the ongoing oscillation, pre-synaptic events will occur after post-synaptic ones, reducing the net synaptic strength (long-term depression, LTD) (*Vossen, Gross & Thut, 2015*). However, the trend for delta activity decrease in the active session was only observed for the cue-P3 (and not in target-P3), which, according to the STDP hypothesis, may suggest that the ERO associated with cue-P3 is lower than the ERO associated with target-P3 (see Fig. 4B).

The results from cue and target-P3 suggest that the preceding identification of the frequency in tACS is an effective way to engage the intended oscillatory activity. This methodology was recently tested in clinical trials with different neuropsychiatric disorders

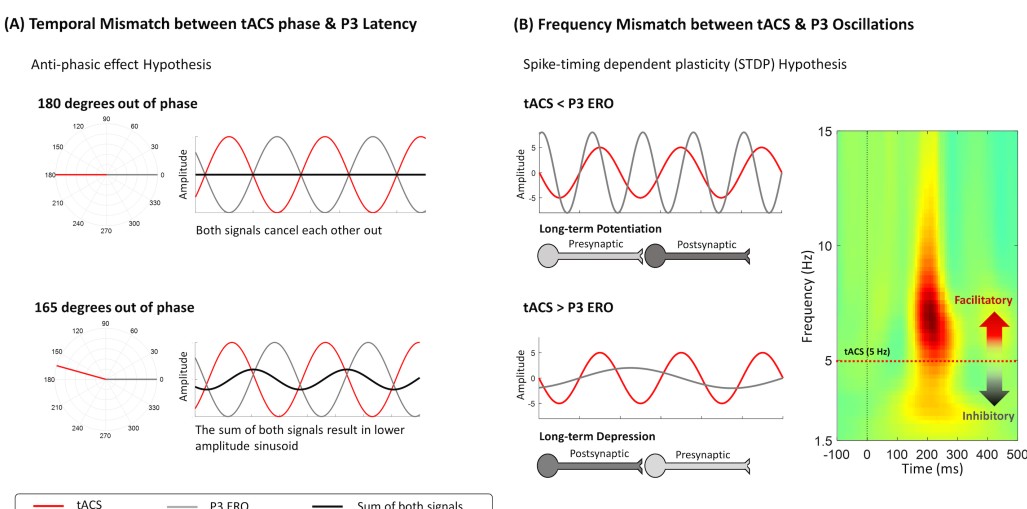

**Figure 4** **Frameworks about mechanisms of action of tACS that might explain the decrease in evoked-delta after tACS.** (A) The anti-phasic effect suggests that the phase of tACS is 180 degrees out of phase with the evoked oscillation. The sum of both signals will be 0, implying that anti-phasic tACS might reduce the amplitude of the evoked oscillation. (B) Another hypothesis is spike-timing dependent plasticity (STDP), suggesting that the strengthening (LTP) or weakening (LTD) of the synapse depends on the difference between the tACS frequency and the ongoing oscillatory activity. Synaptic plasticity is dependent on the timing of presynaptic and postsynaptic events; therefore, it is expected that tACS will have a differential effect on neural oscillations below or above the stimulation frequency. Specifically, oscillatory activity below tACS frequency is inhibited, while oscillations higher than stimulation frequency are facilitated.

(*Huang et al., 2021*; *Riddle et al., 2021*). In recent animal models, the match between the frequency of stimulation and endogenous activity was also highlighted, either because stimulation outside the range of endogenous frequencies may have almost no effect (*Asamoah, Khatoun & Laughlin, 2022*) or even cause a reduction in entrainment at specific frequencies (*Krause et al., 2022*). To further explore this, we performed an additional analysis, in which the temporal and frequency windows for each participant were adjusted (see Fig. S4). The results were consistent with the previous studies, which used the standard P3 time windows and frequency bands. Nonetheless, the effect size observed in the P3 amplitude between both sessions was lower when the time window was adjusted to the individual P3 latency of each participant. This temporal adjustment performed might bias the results because tACS is also capable of modulating P3, as already shown in the literature (*Pahor & Jaušovec, 2018*).

Finally, the current study tried to address some previous methodological limitations. For the online EEG analysis, we had a frequency resolution of 0.25 Hz, instead of 0.5 Hz as reported by previous studies (*Dallmer-Zerbe et al., 2020*; *Popp et al., 2019*). This is of particular interest given that the frequency of tACS is identified during the online analysis, which allows an improvement in stimulation parameters and consequently better modulatory effects. Likewise, both studies mentioned before applied an intensity of 1 mA peak-to-peak, whereas we have decided on a peak-to-peak intensity of 2 mA. This was performed because lower intensities of tACS might not be enough to properly modulate

the intended oscillatory activity (*Johnson et al., 2020*). Furthermore, *Popp et al. (2019)* also pointed out the between-subject design as a caveat in tACS studies; thus, we implemented a within-subject design with two sessions to optimize statistical power. Finally, in terms of behavioral analysis, we used the same reward/punishment system in both sessions, because a recent study found that it was related to the number of premature responses (*Mendes et al., 2024*).

### Limitations and future directions

The sample size calculation was aimed at detecting an effect in P3 amplitude, which resulted in a low sample size to test other hypotheses (*e.g.*, decrease of premature responses). In addition, our study did not replicate exactly all the same methods employed in previous studies. Namely, we performed a within-subjects design and used absolute change instead of relative change. However, the descriptive data suggests an increase in ERO during the active tACS session and a decrease in sham (see Table 1; Fig. 2), as similarly observed by *Dallmer-Zerbe et al. (2020)*. Therefore, the effects on ERO activity should be further tested in studies with larger sample sizes, including both hypotheses raised for the ERO decrease in cue-P3. In specific, future studies should address how the temporal and frequency mismatch between stimulation and EEG endogenous activity (*i.e.*, anti-phasic and STDP hypothesis respectively) can decrease ERO power and consequently reduce ERP amplitude.

   Moreover, this study tested the offline effects of tACS through the absolute change between the blocks performed before and after the stimulation, rather than analyzing the block during the stimulation (*i.e.*, online effects). The literature on tACS suggests offline aftereffects with a duration of at least 30 min (*Neuling, Rach & Herrmann, 2013*) or 70 min (*Kasten, Dowsett & Herrmann, 2016*). On the other hand, other studies have suggested that tACS effects are commonly observed online, rather than offline (*Pozdniakov et al., 2021*). Likewise, P3 latency might change over time (*Devos et al., 2020*), which can represent an obstacle to correct temporal synchronization because we only evaluated endogenous EEG activity for each subject in the first session. However, although we did not observe significant fluctuation of P3 latency during the experimental session (see Table S5), future research should focus on closed-loops stimulations that are dependent on the online endogenous oscillatory activity to optimize tACS effects (*Frohlich & Townsend, 2021*; *Leite et al., 2017*). This can also ultimately allow to increase the precision of tACS on the behavioral outcomes, as the stimulation could also be adjusted based on individual performance. For instance, due to our sample size, we were unable to find differences in terms of release times; however, it seems that active tACS increased it, while sham tACS decreased it. It also seems that while there was an increase in the variability of the responses in the active condition, there was a decrease (when compared to baseline) for the sham condition. Therefore, future studies should take into account these performance-based closed loops, to increase the precision of the effects of tACS in cognitive, as well as behavioral outcomes.

## CONCLUSION

Our findings suggest that is possible to modulate the P3 amplitude through a tACS frequency and phase adjustment to the endogenous activity of each participant. More

specifically, tACS counteracted the expected decrease in P3 amplitude observed in the sham session. Nonetheless, tACS did not lead to significant effects in the ERO during the target-P3 time window, although a significant reduction in evoked-delta was observed in cue-P3 when the analysis was adjusted. This effect might be explained by the differences in the ERO between both P3 components, as well as the fact that tACS was synchronized with the target-P3 instead of the cue-P3. These results highlight the importance of identifying the underlying target activity and the potential of adjusting tACS to that specific activity to optimize its effects.

### Funding

A.J.M. and A.L. are supported by the Portuguese Foundation for Science and Technology and the Portuguese Ministry of Science, Technology and Higher Education through national funds, and co-financed by FEDER through COMPETE2020 under the PT2020 Partnership Agreement (POCI-01-0145-FEDER-007653). J.L. and S.C. are supported by the Portuguese Foundation for Science and Technology and the Portuguese Ministry of Science, through national funds and co-financed by FEDER through COMPETE2020 under the PT2020 Partnership Agreement through PTDC/PSI-ESP/30280/2017 and PTDC/PSI-ESP/29701/2017. The funders had no role in study design, data collection and analysis, decision to publish, or preparation of the manuscript.

### Grant Disclosures

The following grant information was disclosed by the authors:
Portugese Foundation for Science and Technology.
Portugese Ministry of Science, Technology, and Higher Education.
FEDER COMPETE 2020 PT2020 Partnership Agreement: POCI-01-0145-FEDER-07653, PTDC/PSI-ESP/30280/2017, PTDC/PSI-ESP/29701/2017.

### Competing Interests

Jorge Leite is an Academic Editor for PeerJ.

### Author Contributions

- Augusto J. Mendes conceived and designed the experiments, performed the experiments, analyzed the data, prepared figures and/or tables, authored or reviewed drafts of the article, and approved the final draft.
- Alberto Lema analyzed the data, prepared figures and/or tables, and approved the final draft.
- Sandra Carvalho conceived and designed the experiments, analyzed the data, authored or reviewed drafts of the article, and approved the final draft.
- Jorge Leite conceived and designed the experiments, analyzed the data, authored or reviewed drafts of the article, and approved the final draft.

## Human Ethics

The following information was supplied relating to ethical approvals (*i.e.*, approving body and any reference numbers):

The study was approved by the University of Minho ethics committee (CEICVS 057/2021) and was performed in conformity with the Declaration of Helsinki.

## Data Availability

The datasets and code are available at OSF: Mendes, Augusto J. 2024. "Tailoring Transcranial Alternating Current Stimulation Based on Endogenous Event-Related P3 to Modulate Premature Responses: A Feasibility Study." OSF. March 4. doi: 10.17605/OSF.IO/2DGWB.

## Supplemental Information

Supplemental information for this article can be found online at http://dx.doi.org/10.7717/peerj.17144#supplemental-information.

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
