# Peer review of "Tailoring transcranial alternating current stimulation based on endogenous event-related P3 to modulate premature responses: a feasibility study"

_PeerJ, doi:10.7717/peerj.17144_

## Round 0.1 · original submission · Major Revisions

Dear Dr. Leite,

As you will see, the reviewers have a generally positive outlook on your paper but urge you to highlight its limits and consider essential revisions. I invite you to strictly follow their suggestions and resubmit a fully revised version of your manuscript.

Thank you for your submission to PeerJ.

**Language Note:** The review process has identified that the English language must be improved. PeerJ can provide language editing services - please contact us at copyediting@peerj.com for pricing (be sure to provide your manuscript number and title). Alternatively, you should make your own arrangements to improve the language quality and provide details in your response letter. – PeerJ Staff

Reviewer 1 ·

Basic reporting

o This study investigated the feasibility of modulating the P3 amplitude using tailored tACS. The specific aim of this study is to increase the P3 amplitude in a cued premature task.
o The main finding is that tailored tACS modulates the target-P3 but not the cued P3 while not modulating the ERO.
 The English language should be revised to ensure that an international audience can clearly understand the text. There are several typos in the whole text. Some examples where the language could be improved include lines 67 – 70; 89-91; 175-176 – the current phrasing makes comprehension very difficult.
The figures and tables show the results in an appropriate manner

Experimental design

o The authors have introduced a broad research question, which should be more clearly defined. The research area is relevant and meaningful, as modulation of a single ERP component may pave the way for clinical applications. The knowledge gap to be investigated is not clearly identified, and important previous research is neglected (results of Popp et al., (2019). The present study tries to fill the knowledge gap whether it is possible to modulate the P3 amplitude using P3-tailored tACS.
o The authors stated that the study was conducted in accordance with the Declaration of Helsinki. Since the revision in 2008, the declaration states that “Every research study involving human subjects must be registered in a publicly accessible database before recruitment of the first subject.” ( §35 in Version 2013). Was the study registered in an open database? If yes, please provide the necessary information. If not, the statement regarding adherence to the declaration needs rewording.

Validity of the findings

no comment

Additional comments

o A figure showing the electric field in the stimulated area would be beneficial
o The introduction should be revised. Especially the part about the relationship between tACS and the P3, as it sounds to the reader like a mere list of studies without including the relevant findings of the studies referred to.
o Motivation for hypothesis 1 is not established
o The first sentence of the conclusion section seems to be missing a part. (Line 44-46) “Our findings highlight… […]… may result”
o
o The motivation for the study becomes is not entirely clear in the introduction. Why would one want to modulate the P3 at all? What would be a possible benefit of increasing P3 in the waiting impulsivity paradigm?
o Why were the latency and ERO frequency determined only once and not before both sessions? For example, Devos et al. (2020) showed that the reliability of the P3 latency was relatively low. Also, Zamrini et al. (1991) showed that the intrasubject variability was very large although no statistical group difference could be found. Although there are also several studies showing a moderate to high level of reliability in P3 latency, a renewed determination of the parameters in session 2 would have been advantageous. Please argue why you did not make a second determination and please include an analysis showing the test-retest reliability of the stimulation parameters. Therefore, the authors can use the first block of each session and estimate post hoc the P3 ERO and latency.
o Regarding the sample size calculation:
 The authors refer to the effect estimate observed in the study by Dallmer-Zerbe et al. (2020). However, the authors first inferred that the Dallmer-Zerbe study had a within-subject design (line 83) although they had a between-subject design and, furthermore, they did not consider the study by Popp et al. (2019), who performed a study with the same paradigm as the Dallmer-Zerbe et al. study but with healthy participants. Since Popp et al. (2019) did not find a tACS effect in healthy participants, their effect size must also be considered in the sample size calculation, especially, since the authors also conducted their study also with healthy participants.
 The authors missed to mention the effect size measure they used for sample size calculation (d?)
 Please clarify your sample size calculation.
o Since the sample size was calculated based on a relative measure, the authors should first analyze their data in the same way, as Dallmer-Zerbe did (relative rather than absolute changes), to have comparable results.
o Please make sure that the directionality of your hypothesis is reflected in your statistical testing (tails of t-test)
o It appears that one participant was excluded for the electrophysiological results but not for the behavioral results. Please explain, why you did not exclude this participant also for the behavioral analysis.
o In line 278: V(10) – is “V” a typo and the authors meant W(10) or which test was applied there?
o The present study uses a 2 x 2 factorial design with block (pre vs. post) and condition (stim vs. sham) as factors. However, the statistical analysis is limited to paired t-tests comparing the main effect of block (pre vs. post). This analysis is not suitable for testing the hypotheses. I strongly recommend adapting the analysis to include a test of interaction effects to test for differences in condition. From my perspective, it would be beneficial for the authors to include an omnibus test (like an ANOVA – although the data are maybe not normally distributed), if only to increase the statistical power and additionally to show whether there is an interaction between the factors. For example, the authors do not have a test that examines the difference between stimulation and sham.
 The plots of the EROs look promising that there is an interaction between conditions and block. Furthermore, from my point of view, it might be that the stimulation effect gets significant if the sample size is increased.
o The limitations section should also include a paragraph about the difference between the Dallmer-Zerbe study (ADHD patients and between-subjects design) and the present study, and that the authors used a relative measure for sample size calculation, whereas the present study reports only absolute differences. I have read the rationale for reporting absolute changes, but this has also been considered in the sample size calculation.
 The authors found a P3-amplitude effect after verum stimulation but not after sham stimulation (as hypothesized), so sample size is not a limitation in the present study. The authors performed an a priori sample size estimation to circumvent the sample size as being a limitation. Please revise this part of the discussion.
o Did the authors check for outliers?
o All captions need to be completely reworked as they give no information about the effects visible in the plots
o Minor:
 Background in the abstract could give a clearer relationship between the P3 and evoked delta activity
 Use the term participant instead of subject
 No explanation of an oddball-paradigm in introducton
 The study of Dallmer-Zerbe et al. (2020) had no within-subject but a between subject design
 Abbreviation CPRT is not explained (line 128)
 Was the “online” analysis really performed online? I.e., while the participant was performing the task or afterwards, between the baseline block and block 1? Please clarify.
 The wording of the explanation of the three additional analyses is a bit confusing and had to be read several times. Please rephrase this paragraph.
 Why do you draw a neuroConn stimulator in Figure 1 although you used a starstim stimulator?
 Why did the authors choose these electrode positions?
 Which EEG-Channel did you record? Only Cz and Pz?
 In table 1 and table 2 the authors wrote tDCS instead of tACS
 The figure S.M.6 is very stretched
 In caption of figure 1 (C) is mentioned twice

Reviewer 2 ·

Basic reporting

The paper is mostly understandable, but could use some copy-editing and reorganization. In particular, I think the Results section would be easier to follow if the authors indicated the expected outcome. For example, "We observed a significant difference in the target-P3 amplitude [for which the tACS was optimized]" but not the cue P3 which served as a control"...

An OSF repository is mentioned (Line 163), but I did not see a link in the main text, and therefore could not check the data.

Experimental design

The central claim of the paper seems to be that tACS is more effective when tuned to match each subject's P3 timing and spectral content (e.g., "Conclusion: Our fndings highlight the importance of phase synchronization between tACS parameters and the endogenous oscillatory activity.") I am very sympathetic to this idea, but the data presented here do not directly address this claim. The authors compare active, individually tailored tACS against sham stimulation do find some effects, but this is not sufficient to establish that tuning matters; one would need to compare against unoptimized (or deliberately de-tuned) stimulation to establish this. The Cue vs. Target P3 comparisons could help here, but if so the authors would need to demonstrate that the two components are effectively interchangeable (e.g., same amplitude, origin, etc). Otherwise, I think statements about how this work demonstrates the essential need for stimulus optimization ought to be weakened.

More technically, I am also concerned that the frequency tuning may not be effective. The authors appear to be selecting the peak frequency in the CWT of each subject's data (e.g., Fig 1C), which is used to set the frequency of the sine wave used during tACS. However, the ERP does not seem to resemble a sine wave--it rises and falls very rapidly near the peak with a little "plateau" on either side. Moreover, what happens if the P3 latency or spectral content changes as a result of stimulation? This is briefly alluded to in Line 366, but it seems like a big threat to this paradigm.

The task description could be moved from the supplement to the main text.

Validity of the findings

The main positive result is a difference in P3 amplitude after active vs. sham blocks of tACS. However, this is driven by a decrease in during the sham, rather than an increase in the tACS, and this is just mentioned in passing ("counteracted the expected decrease in P3 amplitude"). Could the authors explain why this decrease occurs? The task in the cited paper seems quite different from the present one, and it's not obvious to me that it should be expected.

The negative results are difficult to interpret because the experiments were powered for a very large effect size (1.2; the difference in height between adult men and women is ~1.7). This is based on one cited paper; it would also be helpful to know if other work has found similarly-large effects; it seems like it would be a huge behavioral effect.

In Figure 3, are the data the grand mean (pooled over all subjects) or from a representative subject? It would be helpful to know how consistent these effects are across subjects; this information is mostly absent and it would be nice to see (e.g.,) CIs or error bars.

The blinding and clinical data are not analyzed anywhere in the paper.

Additional comments

Line 384-6 as tACS delivered at lower intensities might not be enough to properly modulate the intended oscillatory activity (Johnson et al., 2020).

The authors might be interested in work by Krause, Vieira, et al. (2022; PLoS Biology) and Asamoah et al. (2022; J Neurosci), who show that low intensity tACS *frequency*-mismatched with the endogenous activity tends to make spiking less oscillatory. This seems like a potential explanation for an online reduction in Cue P3-delta, which (via STDP), could lead to the decrease seen in the subsequent blocks. Alternately, if phase cancellation were the mechanism, one might expect to see the weakest effects in Subjects 1 and 2, who had frequencies of 6-7 Hz vs. the rest in the delta range. Digging into this would also provide a nice rationale for Figure 4, which otherwise just summarizes two hypotheses from the literature.

Reviewer 3 ·

Basic reporting

The manuscript is written in a clear and professional manner throughout.
Sufficient background information was provided on tACS and the theoretical importance of personalising its application to the endogenous activity of the brain, accompanied by sufficient references for further reading.
The structure of the article conformed to the standard format, figures are clear and of a sufficient resolution. The legends for figures 2 and 3 appear to be incorrect, or at least a little confusing. It may be more clear to list the subplot letter before the relevant information, e.g., “(B) The event related potentials in the target period” for figure 2. The legends also don’t quite describe the plot shown, for example the figure 2 legend says “the event related potentials (B) in the time-window of interest (represented in the gray area and dashed lines: 250 - 450 ms)”, but in fact a larger time window is shown. Some added clarity or accuracy here would be helpful to understand the results.
The raw data did not appear to be available. Thought this was unclear to me.
The manuscript is coherent and self-contained.

Experimental design

The research presented appears consistent with the aims and scope of PeerJ.
The experimental design is in line with the research question, if perhaps sub-optimal. Personally, I would find it more convincing if in-phase tACS was contrasted with anti-phase tACS, since using the current data one cannot distinguish effect of personalised tACS from that of tACS generally. Though, this is issue, of insufficient control conditions, is very common in the field of personalised and closed-loop neuromodulation. This should be addressed in the limitations section.
The methods are outlined in sufficient detail for replication of the experiment and analysis.

Validity of the findings

Analyses seem to be sound, robust.
Conclusions are not definitively supported by these data (see anti-phase control above), but they are made in a sufficiently conservative manner so as not be a big issue. Future directions should mention this important control.

Additional comments

1. Lack of active (anti-phase) control must be addressed
2. Figures 2 and 3 would be more easily readable and understandable to me, personally, if those data which are statistically compared were plotted together. i.e., the pre-post ERP’s are contrasted, it would be nice to see these plotted, ideally together.
3. Figure 4, and much of the associated discussion about differences in the cue period, goes beyond the data to an extent which I found unhelpful. This difference in the power spectra during the cue period may be easier to interpret if the pre-tACS traces are plotted for active and sham stimulation together. For instance, it is unclear from the plots, and statistics, if active- and sham- conditions were different in the pre-tACS period. The difference in their changes may result from this, in some regression to the mean.
4. Typos:
• Line 64: “outlast stimulation for few weeks” should be “outlast stimulation for a few weeks”
• Line 68: “componente” should be “components”
• Line 115: “Out” should be “Our”
• Line 207: “and then At next, re-referenced” may mean “before re-referencing”

---

## Round 0.2 · Major Revisions

While your earlier revision improved the manuscript, I invite your to thoroughly consider the remaining reviewer's suggestions.

Reviewer 1 ·

Basic reporting

I am grateful that the authors have already implemented many points very well as part of the review process. However, there are a few open issues I would like to be addressed.

1. Since the last assessment more relevant studies in the field have been published, on the effect of P3-tailored tACS: (Kannen et al., 2022; Boetzel et al., 2023). These should also be included, as they show further important results on the feasibility of modulating the P3 amplitude and EROs with P3-tailored tACS.

Experimental design

2. I feel that the hypothesis section of the introduction would benefit from an overhaul: Currently there is no clear distinction between broad research goals and specific (testable!) hypotheses. The effect of entrainment is just a presumed method of action, not an actual hypothesis. Otherwise it would require testing via a selected measure of phase-analysis between brain rhythm and stimulation. The main hypothesis would therefore be: H1: Significant increase between stim and sham for target-P3. H2: No significant increase between sham and stim for Cue-P3, H3: Significant correlation between target-P3 and amount of premature responses. The H2 is also not entirely convincing to me. I understand this reasoning for on-line effects during stimulation. But, as Analysis of the P3 effects is based on the post-stimulation data, any effects there would be expected to be reliant on plastic changes of the band-power. These ban-power changes should affect all P3s, irrespective of the phase-differences to the no longer active stimulation (as we that any eventual entrainment end after the end of stimulation (Strüber et al., 2015; Vossen et al., 2015)).

3. Did the authors control for drifts between the tACS and the visual stimulation and if so, how? From the current manuscript, it sounds like the systems were only synchronized in the beginning of the stimulation. As the clocks of different systems eventually drift apart, was there a control to maintain synchronization for the entire duration or a check on the magnitude of drift?

Validity of the findings

4. Although I think it's a really good approach that the authors wanted to replicate the results of Dallmer-Zerbe et al., (2020), and have therefore followed the analysis steps in this paper, I would recommend a further analysis of the EROs. An additional analysis of the ERO around stimulation frequency ± 1 Hz would be beneficial to analyze the tACS after effect. This is because of the assumption, that if tACS after effects are caused by entrainment, it is thought to be most effective when the stimulated frequency is at or close to the endogenous frequency effects (Arnold tongue). Furthermore, the majority of studies that report aftereffects have outlined frequency-specific offline effects of stimulation e.g., (Zaehle et al., 2010; Neuling et al., 2013; Kasten et al., 2016). Thus, the power of the individual delta/theta frequency of the post block should be investigated in the range of the individual stimulation frequency band ± 1 Hz and be compared to the pre-power. If I understood it correctly, the authors did an individualized ERO analysis for each participant´s individual frequency ± 3 Hz. This is a wide range of frequencies and results in many different amounts of data points for the average power value calculation. Thus, I would recommend smaller frequency bands (± 1 Hz) to keep the size of the analyzed windows as similar as possible. Furthermore, a difference plot between pre- and post-block would improve the presentation of the results.

5. Is there a reason for the asymmetric SDs of RT for the feedback? Please explain the reasoning shortly in the manuscript.

6. The authors analyzed the cue-P3, however, the ERP do not show a P3 component, as it is expected from a cue stimulus. Thus, the authors should rephrase the parts with the cue P3. Maybe they could just mention a modulation of the ERP, but not of the P3.

7. Furthermore, I did not get the point, why the authors matched the tACS to the target-P3 and not the cue-“P3” if they wanted to investigate the effect of the tACS on the impulsivity.

8. Additionally, the authors argue that the Cue-P3 might be decreased because of an antiphasic tACS. However, the time interval between the cue and target was jittered between 1000 to 1500 ms. Depending on the stimulation frequency and the time-interval, such a match with a trough and the cue-P3 is unlikely and might happen only by chance. Here, a presentation of the phase of the tACS during the cue-“P3” could help to support their suggestion that the ERP component descriptively decreased because of an antiphasic tACS.

Additional comments

Minor:

Please correct following typo:

Line 138 „Afterward,“ should be „Afterwards,”

Please mention, that the study by Mendes et al., 2022 is a pre-print.

If the statistical tests were performed two-sided, the alpha level to be assumed is 0.025 and not 0.05.

In the whole manuscript, it is necessary to check the citation style. Some citations are mentioned several times (e.g. line 261ff, 271 ff. etc.)

The resolution of the figures has to be improved, as it is impossible to read some of the axis labels.

Reviewer 2 ·

Basic reporting

Basic reporting is sufficient

Experimental design

Experimental design is pretty well described but the rationale for the phase-tuning should be elaborated. See PDF.

Validity of the findings

Conclusions have been appropriately narrowed to the feasibility in most place (see PDF for notes). As the other Reviewer notes, Figure 4 still does go a bit beyond the data but is fairly clearly described as speculation.

Additional comments

See attached PDF.

Annotated reviews are not available for download in order to protect the identity of reviewers who chose to remain anonymous.

---

## Round 0.3 · accepted · Accept

You have addressed all of the reviewers' substantial comments, and thus I deem the paper ready for publication.

Reviewer 1 ·

Basic reporting

no comment

Experimental design

no comment

Validity of the findings

no comment